# How Do Phosphorus Compounds with Different Valence States Affect the Flame Retardancy of PET?

**DOI:** 10.3390/polym15081917

**Published:** 2023-04-17

**Authors:** Siheng Zhao, Bo Xu, Hao Shan, Qinglei Zhang, Xiangdong Wang

**Affiliations:** 1School of Chemistry and Materials Engineering, Beijing Technology and Business University, Beijing 100048, China; 2Beijing Key Laboratory of Quality Evaluation Technology for Hygiene and Safety of Plastics, Beijing 100048, China; 3China Light Industry Engineering Technology Research Center of Advanced Flame Retardants, Beijing 100048, China

**Keywords:** polyethylene terephthalate (PET), flame retardant, valence states, phosphate

## Abstract

This work investigated the effect of different valence states of phosphorus-containing compounds on thermal decomposition and flame retardancy of polyethylene terephthalate (PET). Three polyphosphates—PBPP with +3-valence P, PBDP with +5-valence P and PBPDP with both +3/+5-valence P—were synthesized. The combustion behaviors of flame-retardant PET were studied and the structure–property relationships between the phosphorus-based structures with different valence states and flame-retardant properties were further explored. It was found that phosphorus valence states significantly affected the flame-retardant modes of action of polyphosphate in PET. For the phosphorus structures with +3-valence, more phosphorus-containing fragments were released in the gas phase, inhibiting polymer chain decomposition reactions; by contrast, those with +5-valence phosphorus retained more P in the condensed phase, promoting the formation of more P-rich char layers. It is worth noting that the polyphosphate containing both +3/+5-valence phosphorous tended to combine the advantage of phosphorus structures with two valence states and balance the flame-retardant effect in the gas phase and condensed phase. These results contribute to guiding the design of specified phosphorus-based structures of flame-retardant compounds in polymer materials.

## 1. Introduction

Poly (ethylene terephthalate) (PET) is used in all aspects of life because of its excellent corrosion resistance, ease of processing, and excellent mechanical properties [1,2,3]. However, its highly flammable burning process releases a lot of smoke, which may endanger humans [4,5,6,7,8,9].

Therefore, many scholars have devoted themselves to developing PET composites with excellent flame-retardant properties. Phosphorus-based flame retardant was considered an excellent alternative to halogen-containing flame retardant because of its low toxicity and high flame-retardant efficiency. Kilinc [10] used different phosphorus-based flame retardants as additives to improve the flame retardancy of PET and found that the LOI of the flame-retardant composites increased from 21% of neat PET, up to 36% with the addition of 5% BP and 5% triphenyl phosphate to the matrix. Fang [11] synthesized a novel flame retardant (DOPO-TPN) based on phosphaphenanthrene and cyclotriphosphazene, and the results showed that 5% DOPO-TPN achieved a LOI value of 34% and UL-94 V-0 rating.

At the same time, in order to ensure that the mechanical properties of the compound will not be greatly affected during processing, more and more PFRs are receiving attention, and polyphosphonate plays a major role [12]. Some researchers have proved that phosphorus-containing flame retardants can play a flame-retardant role in the gas phase and the condensed phase. In the gas phase, phosphates mainly exerted a flame-retardant effect by releasing phosphorus-containing radicals to play quenching roles [13]. In terms of the condensed phase, phosphorus flame retardants mainly play a role in promoting the char layer with high viscosity, thus preventing the release of volatile fuels and heat from the substrate matrix.

However, the change of structure leads to differences in the valence state of phosphorus, which further affects the proportion of flame-retardant mechanisms in the condensed and gas phases [14,15,16]. Therefore, studying the effects of flame retardants with different phosphorus valence states on the flame-retardant properties of PET composites helps to guide the design and synthesis of highly efficient flame retardants.

In this work, PBPP, PBPDP and PBDP with different phosphorus valence states were synthesized and used in PET to analyze the effect of different phosphorus valence states on the flame-retardant mechanism.

## 2. Experimental

### 2.1. Materials

Phenylphosphonic Dichloride (PPD), Petroleum ether and Dichloromethane were provided by Beijing Innochem Technology Co., Ltd. (Beijing, China). 4,4′-isopropylidenediphenol (BPA), and Phenyl dichlorophosphate (PCDP) were provided by Shanghai Aladdin Biochemical Technology Co., Ltd. (Shanghai, China). Triethylamine was provided by Tianjin FuChen chemical Co., Ltd. PET (FSPG, Tianjin, China) with M_n_ about 20,000 g/mol.

### 2.2. Synthesis of Polyphosphates

The synthesis of flame retardants was carried out by changing the reaction of different phosphorus-containing dichloride and BPA (0.05 mol). The detail process of the reaction is shown in Figure 1. In a three-necked and round-bottomed flask with a stir, phosphorus-containing dichloride (PPD 0.05 mol or PPD/PDCP 0.025 mol/0.025 mol or PDCP 0.05 mol) and Triethylamine (0.05 mol) were added into the Dichloromethane (50 mL) at room temperature first, until completely dissolved, then dichloro compounds (0.05 mol) were added dropwise into the flask in 1 h. After dripping, the mixed solution reacted for 4 h at 40 °C under the protection of N_2_. At the end of the reaction, Petroleum ether was used for reprecipitation to get the final product. The synthesis process of the three polyphosphates was similar, with yields of 87%, 81% and 86%, respectively.

### 2.3. Preparation of PET Composites

Before processing, PET and flame retardants were dried in a vacuum oven at 160 °C for 12 h. The test sample was processed in the torque rheometer at 270 °C, and then processed into splines by a tablet press according to the test standard. The addition of flame retardants in all PET composites was 8 wt.%, and the dimensions of the samples were 100 mm × 100 mm × 30 mm.

### 2.4. Characterization

Cone calorimeter test was carried out by FTT0007 (Fire Testing Technology, East Grinstead, UK) according to ISO 5660:2015 standard.

Thermogravimetric analysis (TGA) was carried out by a STA 8000 thermogravimetric analyzer (Perkin Elmer, Waltham, MA, USA). The temperature range was 50 °C to 700 °C and the heating rate was 20 °C/min.

A thermogravimetry-Fourier transform infrared (TG-FTIR) test was carried out on a STA 8000 simultaneous thermal analyzer (Perkin Elmer, Waltham, MA, USA). The temperature range was 50 °C to 700 °C and the heating rate was 20 °C/min.

The microscopic pictures of carbon residues were taken by a Phenom Prox scanning electron microscope (SEM) (Phenom World, Eindhoven, Holland).

The Fourier transform infrared (FTIR) spectroscopy was carried out by a Nicolet iN10MX-type spectrometer (Nicolet, USA) in the range of 4000–500 cm^−1^.

The Hot-Stage Fourier transform infrared was performed on a LNP96-S/iS50 spectrometer (Linkam Scientific Instruments, Redhill, UK).

The pyrolysis of flame retardants was carried out by GCMS-QP 2010 Plus GC–MS with a PYR-4A pyrolyzer (Shimadzu, Kyoto, Japan).

## 3. Results and Discussion

### 3.1. Chemical Structure and Pyrolysis Behavior of Flame Retardants

Figure 1(a1,b1,c1) showed the FTIR spectrum of PBPP, PBPDP and PBDP. The absorption band at 1270 cm^−1^ was attributed to the adsorption of P=O. Two absorption bands at 1200 and 958 cm^−1^, corresponding to the vibration of P-O-C bonds, can also be observed clearly. The peak at 3061 cm^−1^ corresponded to the Ar-H. Moreover, absorption bands at 1364 and 1408 cm^−1^ corresponded to the methyl bending vibrations [14].

The structural characterization of PBPP, PBPDP and PBDP is shown in Figure 1. In the ^1^H NMR spectrum of PBPP, peaks at 7.2–8.0 ppm were from the aromatic ring peaks of a phosphorous-containing side group, and signals at 7.0–7.2 ppm corresponded to the aromatic ring peaks of bisphenol A. The ^1^H NMR spectra of PBDP were different from those of PBPP. The aromatic ring peaks of a phosphorous-containing side group in PBDP appeared in 7.0–7.5 ppm. Additionally, the ^1^H NMR spectrum of PBPDP was a combination of PBPP and PBDP. Meanwhile, in the ^31^P NMR spectrum of PBPDP, there were two peaks, which corresponded to the P-O-Ar (12.64 ppm) and P-Ar (16.33 ppm) [17]. As indicated by the data of FTIR, ^1^H SSNMR and ^31^P SSNMR, three flame retardants were successfully synthesized.

### 3.2. Thermal Stability Analysis

The thermal stability of polymers is an important property in the process. TGA was used to test the thermal stability of samples in nitrogen and air with the results shown in Figure 2 and Table 1. With +5-valence phosphorus increasing from PPBP to PPDP, the initial decomposition temperature rose by the maximum value of about 40 °C. This was because +5-valence phosphorus existed in the form of P-O-C bands in compounds, which were less stable than P-C bonds, as the main existence form of +3-valence phosphorus [18]. However, there was no significant difference for Y_c_ (residual yields at 700 °C), which indicated that the carbon formation ability of polyphosphate was hardly affected by the valence states of phosphorus.

The thermal stability of pure PET, PBDP/PET, PBPDP/PET and PBDP/PET composites under N_2_ atmosphere was also studied (Figure 2 and Table 1). All polyphosphonates caused the earlier decomposition of PET composites in N_2_ atmospheres, due to the unstable P-O-C bonds in polyphosphate [19,20,21]. However, compared with pure PET, the Y_c_ of three flame-retardant PET composites all increased, which indicated that the phosphate improved the charring capacity of PET composites. This phenomenon may be because of the transesterification between the polyphosphonate and matrix chains during the heating process and because the weak P-O-C bond led to the early decomposition of PET composites, while also further increased the quantity and quality of residual carbon. In addition, the theoretical carbon residue was calculated according to Equation (1) and listed in Table 1. An obvious synergistic charring effect between polyphosphates and matrix was found, especially for PPDP with +5-valence phosphorus. Further, although there was almost no difference in Y_c_ values of three polyphosphates, there was a big difference in the synergistic charring ability with PET. As shown in Table 1, PBDP containing +5-valence phosphorus obtained the most obvious synergistic charring ability in PET with the largest Y_c_ value of 23.7 wt.% at 700 °C, but this ability decreased as the partial +5-valence phosphorus was replaced by +3-valence ones, which indicated that +5-valence phosphorus was more likely to react with the matrix through transesterification with ester chains due to its more P-O-C bonds, forming P-containing crosslinking structures in the condensed phase, thus improving the carbon formation ability of PBDP composite. Moreover, the R_max_ (the maximum rate of weight loss) of PET composites showed an opposite trend to that of the flame retardants. That may be because the polyphosphate containing more unstable P-O-C bonds perhaps decomposed earlier, so giving a lower R_max_. On the other hand, after polyphosphates were introduced to PET, +5-valence phosphorus acted more in the form of phosphoric acid or analogues in the condensed phase during the decomposition process, which promoted the dehydration and transesterification of PET chains and accelerated the formation of char layers.
(1)$$W_{calculation}=W_{flame\;retardant}\times R_{flame\;retardant}+W_{PET}\times R_{PET}$$
*W*: The content of the additions in the composites.*R*: The residue result after TGA test.

### 3.3. Force Combustion

The cone calorimetry test simulates a real fire by forcing the combustion which was used to analyze the heat and smoke release of PET composites. The results are shown in Figure 3 and Table 2. Pure PET exhibited a high PHRR value of 1240 kW/m^2^, while the values decreased to 603.6 kW/m^2^, 468.9 kW/m^2^ and 408.4 kW/m^2^ for PBDP/PET, PBPDP/PET and PBDP/PET, respectively. It was not difficult to find that with the increase of +5-valence phosphorus content, PHRR of composites decreased gradually, which might be due to the influence of different phosphorus-containing structures on the flame-retardant mechanism, more specifically on the quality of residual carbon in the condensed phase. THR showed the total heat release from the material during combustion and the trend of THR was similar to that of HRR (Figure 3b), the values of which gradually decreased with the increase of the +5-valence phosphorus content.

Fire growth index (FGI=PHRR/t-PHRR) was used to evaluated the fire hazards [22]. The FGI of pure PET was 9.5 kW/(m^2^·s). As PBPP was added, the FGI value of PBPP/PET decreased to 6.0 kW/(m^2^·s), which indicated that PBPP can effectively reduce the risk of fire. What was more, PBPDP/PET (4.7 kW/(m^2^·s)) and PBDP/PET (4.8 kW/(m^2^·s)) exhibited much lower FGI values, which indicated that flame retardants with +5-valence phosphorus exhibited higher fire safety performance in real fires [23].

The smoke release during combustion is also considered an important influence in fire safety. In general, more char residue means that more decomposition fragments are retained in the condensed phase rather than released to the gas phase. As shown in Figure 3d, the higher the carbon residues, the lower the smoke release; the addition of PBDP PBPDP and PBDP all significantly reduced the TSR values, and the suppression effect was more obvious with the increase of +5-valence phosphorus content. Moreover, compared with neat PET, the decrease of av-CO_2_Y values in all PET compounds was due to the suppression of PET composite combustion. In addition, compared with the av-COY value of PET, only that of PBPP/PET rose [24], which proved that +3-valence phosphorus can release more phosphorus-containing species to exert a gas phase flame-retardant effect, thus causing incomplete combustion of the PET matrix and increasing the av-COY value.

It should be noted that there seems to be some interaction between the two different valence states of phosphorus. The theoretical cone data of PBPDP/PET were calculated according to the proportion of two different phosphorus elements in the ^31^P NMR spectrum of PBPDP. As shown in Table 2, the actual cone test results of PBPDP/PET were all better than the theoretical calculation results in both heat and smoke suppression, which indicated that two-valence phosphorus elements played a synergistic effect when used together, thus improving the flame-retardant efficiency.

Additionally, three important flame-retardant effects (flame-inhibition effect, charring effect, barrier and protective effect) were calculated by Equations (1)–(3) according to the cone test values to evaluate the main flame-retardant modes of action. The results are listed in Table 3.

As shown in Table 3, PBDP/PET possessed a better flame inhibition effect than PBPP/PET, which indicated that the flame-retardant efficiency of +5-valence phosphorus in the condensed phase was higher than that of +3-valence phosphorus in the gas phase. Moreover, the better charring effect and barrier and protective effect of PBDP/PET confirmed that +5-valence phosphorus was better than +3-valence phosphorus in improving the carbon-forming ability of composites and the quality of residual carbon, which not only promoted the dehydration and carbonation of the matrix, but also reacted with the decomposition products to form a char layer containing rich P-O-C bonds, thus presenting the better charring effect and barrier and protective effect.
(2)$$Flame\;inhibitioneffect=(1-\frac{EHC_{fr}}{EHC_{pure}})\times 100\%$$
(3)$$Charring\;effect=(1-\frac{TML_{fr}}{TML_{pure}})\times 100\%$$
(4)$$Barrier\;and\;protective\;effect=(1-\frac{\frac{PHRR_{fr}}{PHRR_{pure}}}{\frac{THR_{fr}}{THR_{pure}}})\times 100\%$$

### 3.4. Physical and Chemical Analysis of Char Residue

The macro- and micro-surface morphology of residues after the cone calorimetry test were investigated. As shown in Figure 4, pure PET presented a porous and poor protective effect of the char layer. Macroscopically, the char residues of PET composites with phosphate/phosphonate were significantly increased. Additionally, the residues of PBPP/PET showed a significant increase in voids, although there were still some voids that existed to allow heat release. As shown in Figure 4(c3), it was found that although the number of residual carbon holes of PBPDP/PET char layer did not decrease, these holes were adhered by a film, because the presence of the +5-valence phosphorus promoted the forming of a high-viscosity P-containing char layer, thus preventing the release of volatile fuels and heat from the substrate matrix. Further, the char layer of PBDP/PET with +5-valence phosphorus was denser and almost free of holes on the surface. Combined with HRR curve, it was shown that +5-valence phosphorus can more effectively improve the quality of the residual carbon in the condensed phase, thus achieving a better barrier and protective effect to prevent the heat/fuel transfer.

In addition, the residual carbon after the cone volume test was analyzed elementally by X-ray photoelectron spectroscopy (XPS). Element contents, the reserved and released phosphorus (P) contents were calculated and shown in Table 4. As seen, for PBPP/PET, more than half of the total P (60.7%) was released into the gas phase as P-containing free radicals; most of the P (92.7%) in PBDP/PET was reserved in the solid phase. This indicated that +3-valence phosphorus was mainly released into the gas phase as P-containing free radicals and played the role of free-radical quenching, while P-containing compounds or species, such as phosphoric acid pyrophosphoric acid, polyphosphoric acid and so on, formed by +5-valence phosphorus, more effectively participated in the charring reactions with the decomposition products to form P-rich residues. Moreover, PBPDP balanced the distribution of phosphorus in the gas (39.6% P) and condensed phases (60.4%), which not only acted as a quenching effect in the gas phase, but also improved the quality of residual carbon in the condensed phase, thus providing better barrier protection.

Figure 5 showed the RT-FTIR of the condensed phase products for PET samples. From Figure 5a, the characteristic absorbance peaks of pure PET mainly included hydrocarbon at 1280–1470 cm^−1^, -C=O at 1762 cm^−1^ and a C-H bond at 2956 cm^−1^, and the relative absorption strength of these peaks started to decrease at 350 °C and almost disappeared at 400 °C. In Figure 5b–d, these characteristic peaks disappeared at nearly 450 °C, indicating that the addition of polyphosphonates can facilitate the dehydration of a matrix into carbon with a high-temperature thermostability. Compared with PBPP/PET, a new peak appeared at 956 cm^−1^, corresponding to the P-O-Ar stretching vibration in PBDP/PET and PBPDP/PET, and the peak intensity increased obviously with the increase of +5-valence phosphorus content [25,26]. This phenomenon indicated that +5-valence phosphorus participated mainly in the charring reactions, which not only promoted the dehydration and carbonation of composites, but also reacted with the decomposition products to form more char layers containing rich P-O-Ar cross-linked bonds, which blocked the release of heat and combustible volatile products during the combustion process. It should be noted that the absorbance of hydrocarbon in PBDP was significantly stronger than that in PBPP, which indicated that +5-valence phosphorus can retain more C-H compounds in the condensed phase, rather than being released into the air as fuels.

### 3.5. Gas-Phase Products Analysis of Flame Retardants and PET Composites

TG-FTIR tests were performed to analyze the gas products of the PET composites, and the results are shown in Figure 6. As shown in Figure 6a–c, the FTIR peaks of pyrolysis for all polyphosphates at different temperatures were relatively similar, such as CO_2_ peaks at 2360 cm^−1^, CO peaks at 2180 cm^−1^, C-C bands at 1598 and 1508 cm^−1^, and phosphorus-containing species peaks at 1255, 1190 and 962 cm^−1^. However, the variation of peak intensity with temperature was obviously different. PBPP showed no obvious absorption peak before 525 °C, and reached the peak rapidly at 550 °C, while PBDP began to decompose in advance at 475 °C, and the absorption peak gradually increased to the peak, which was as same as the TGA curves. Moreover, it was observed that the absorption peak of P-O-C in PBPP was significantly stronger than that in PBPDP and PBDP, which indicated that the polyphosphate with +3-valence phosphorus can release more phosphorus-containing species with a quenching effect, thus mainly playing a gas-phase flame-retardant effect.

Figure 6d shows the mainly pyrolysis products of phosphate from TGA-MS test. The types of pyrolysis fragments of PBPP, PBPDP and PBDP are similar to each other and the possible pyrolysis route is exhibited in Figure 6e. The flame retardants were first pyrolyzed and broken at the P-O bonds to generate two parts (phosphate-containing structure and BPA) due to the unstable P-O bonds. The pyrolysis fragments at *m*/*z* = 47 (PO•) and 63 (PO_2_•) were mainly attributed to the cleavage of phosphorus-containing compounds. Additionally, due to the different structure of phosphate, R can be divided into a benzene ring (*m*/*z* = 78) and phenol (*m*/*z* = 94) [27]. On the other hand, the BPA part decomposed into a series of aromatic compounds.

The changes of gas-phase products were also detected by TGA-FTIR, and the results are shown in Figure 7a–d. At the same time, in order to further explore the effect of different phosphorus valence states on the gas-phase products of PET composites, Figure 7e,f shows the absorbance change of C=O and P-O-C at different temperatures. The degradation products of PET included aliphatic ether at 1020–1180 cm^−1^, hydrocarbon at 1280–1470 cm^−1^, CO_2_ at 2373–2354 cm^−1^, -C=O at 1762 cm^−1^ and CO at 2233–2144 cm^−1^. Moreover, most of the gaseous products of the three PET composites were the same as those of PET. It should be noted that a new absorbance peak of phosphorus-containing structures (P-O-C) appeared in TGA-FTIR of PBPP/PET, while with the decrease of 3-phosphorus content in flame retardants the absorbance peaks gradually decreased, and eventually disappeared in PBDP/PET. In addition, the changes of P-O-C release intensity in Figure 7e further proved this trend with phosphorus valence states. This phenomenon proved that +3-valence phosphorus can release more PO and PO_2_ free radicals in the gas phase, so as to play a stronger role in gas-phase action [24,28]. During the combustion, C=O (1762 cm^−1^) and hydrocarbon (1280–1470 cm^−1^) fragments were usually released into the gas phase as volatile combustible fuels to sustain the combustion process [29]. From Figure 7, we can find that the absorption peaks of hydrocarbon and C=O significantly decreased with the increase of +5-valence phosphorus content in polyphosphonates; Figure 7f more clearly shows this trend, which indicated that, due to the high charring effect of +5-valence phosphorus, more flammable hydrocarbons were retained in the condensed phase and reduced the release of fuels, thus suppressing the combustion of composites.

Based on the analysis of gas- and condensed-phase products of flame retardants and composites, the effects of different phosphorus valence states on the flame-retardant mechanism of PET are summarized in Figure 8. Composites containing +3-valence released more P-containing free radicals in the gas phase, which exerted a quenching effect and terminated the combustion reaction, while the one containing +5-valence showed higher charring ability, which was more conducive to the formation and stability of condensed-phase chars, thus inhibiting the heat transfer and preventing the release of gaseous products. In addition, when +3/+5-valence phosphorus was used together, the flame-retardancy of the complex in the gas and condensed phases tended to combine the advantages of phosphorus in these two valence states.

## 4. Conclusions

In this work, three polyphosphates with different phosphorus valences were synthesized to investigate the effect of different phosphorus valences on their flame-retardant mechanism in PET. The results showed that the valence structure affected the proportion of phosphorus exerting flame-retardant roles in the gas/condensed phases. PBPP with +3-valence phosphorus played more pivotal roles in the gas phase, which generated more P-containing free radicals during the combustion process to eliminate H· and HO· free radicals, thereby inhibiting the continuous progress of combustion reactions. On the other hand, PBDP with +5-valence phosphorus was more likely to react with the matrix through transesterification with ester chains due to its greater P-O-C bonds, forming P-containing crosslinking structures in the condensed phase, thus improving the amount and quality of char residue. It should be noted that the flame-retardant test results of PBPDP containing both +3- and +5-valence phosphorus were better than the theoretical calculated values, which indicated that there was a synergistic flame-retardant effect between the two different valence states of phosphorus which tended to combine the advantages of phosphorus in these two valence states.

## Data Availability

No new data were created or analyzed in this study. Data sharing is not applicable to this article.

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
