# Peer review of "How Do Phosphorus Compounds with Different Valence States Affect the Flame Retardancy of PET?"

_polymers, 2023, doi:10.3390/polym15081917_

Round 1

Reviewer 1 Report

This paper compares the flame retardant efficiency of phosphorous flame retardant additives with different valence states. The paper is good, the flame retardancy experiments are all adequate. The main issue with the paper is the poor level of English. The following issues need to be addressed:

1)  The level of English needs to be improved. There are many grammatically badly constructed sentences.

2) Maybe the title can be changed into a more intriguing one such as "How do phosphorus compounds with different valence states affect the flame retardancy of PET?". This is just a suggestion.

3) Section 2.2 should be revised.  ".......by changing the reaction of differ-
ent dihydroxyl compounds and BPA(0.05mol)." Different dihydroxy compounds should be phosphorous compounds I guess? "............, dihydroxyl
compound (PPD 0.05mol or PPD/PDCP 0.025mol/0.025mol or PDCP 0.05mol)."  Simillarly, dihydroxyl should be corrected.

4) Scheme 1. The polymer structures are wrong. The repeating units need to contain one more oxygen for PBDP and PBPP For PBPDP, add another oxygen as well as another BPA unit.

5) Section 3.1: It is not good to say neopentane structure. Instead, you can state that the bands (by the way use FTIR vibration bands or just bands instead of peaks) are associated with the methyl bending vibrations.

6) Figure 1, 5, 6, and 7 should be made more legible. Increase the font of the letters.

7) Page 5: Above Table 2 (which should be Table 1), there is a formula (1) but it is not mentioned or explained.

8) Table 2 on page 7: Control the units of the cone test results. Write m2 as m2.

9) Add the following references which are about the effect of different valence states or phosphorous flame retardants:

- Lin, F., Lin, H., Ke, J., Liu, J., Bai, X., & Chen, D. (2019). Preparation of reactive and additive flame retardant with different oxidation state of phosphorus on the thermal and flammability of thermoplastic polyurethane. Compos. Mater, 3, 43.

-Mariappan, T., Zhou, Y., Hao, J., & Wilkie, C. A. (2013). Influence of oxidation state of phosphorus on the thermal and flammability of polyurea and epoxy resin. European polymer journal, 49(10), 3171-3180.

-Cakmakci, E., & Kahraman, M. V. (2015). Boron/Phosphorus-containing flame-retardant photocurable coatings. Photocured Mater, 150-187.

-Denis, M., Coste, G., Sonnier, R., Caillol, S., & Negrell, C. (2023). Influence of Phosphorus Structures and Their Oxidation States on Flame-Retardant Properties of Polyhydroxyurethanes. Molecules, 28(2), 611.

-Braun, U., Balabanovich, A. I., Schartel, B., Knoll, U., Artner, J., Ciesielski, M., ... & Pospiech, D. (2006). Influence of the oxidation state of phosphorus on the decomposition and fire behaviour of flame-retarded epoxy resin composites. Polymer, 47(26), 8495-8508.

 -Ozukanar, O., Cakmakci, E., Daglar, O., Durmaz, H., & Kumbaraci, V. (2022). A double‐click strategy for the synthesis of P and N‐containing hydrolytically stable reactive flame retardant for photocurable networks. Journal of Applied Polymer Science, 139(35), e52837.

10) UL-94 is mentioned in the methods section but it is not mentioned in the text.

11) Be careful with the use of the spacing. Sometimes there is a space between a number and a unit (or a word) sometimes there is not.

Reviewer 2 Report

The subject of the manuscript focused on an exploration of the relationship between phosphorus-containing structure in and flame-retardant mechanism of poly-phosphate in PET is in good relevance with the scope of POLYMERS.

The introduction very shortly presents the issues related to phosphorus-based flame retardants applied in PE composites. Some publications were cited, however, the results obtained in the cited studies were not discussed in detail. Therefore, the Introduction part requires a more precise presentation of the obtained results described in the literature.

In the section on Materials and Methods, more detailed information on how to prepare test specimens in point 2.3 should be given. The fragment "then processed into splines by a tablet press according to the test standard" should be supplemented with a description of the test standard.

The obtained results are presented and discussed in a correct manner and the conclusions are a good summary of the obtained results.
